

# Filamentous cyanobacteria preserved in masses of fungal hyphae from the Triassic of Antarctica

Carla J. Harper[1,2,3,*], Edith L. Taylor[2] and Michael Krings[1,2,4,*]

[1] SNSB-Bayerische Staatssammlung für Paläontologie und Geologie, Munich, Germany
[2] Department of Ecology and Evolutionary Biology, and Biodiversity Institute and Natural History Museum, University of Kansas, Lawrence, KS, United States of America
[3] Botany Department, Trinity College Dublin, Dublin, Ireland
[4] Department für Geo- und Umweltwissenschaften, Paläontologie und Geobiologie, Ludwig-Maximilians-Universität, Munich, Germany
[*] These authors contributed equally to this work.

## ABSTRACT

Permineralized peat from the central Transantarctic Mountains of Antarctica has provided a wealth of information on plant and fungal diversity in Middle Triassic high-latitude forest paleoecosystems; however, there are no reports as yet of algae or cyanobacteria. The first record of a fossil filamentous cyanobacterium in this peat consists of wide, uniseriate trichomes composed of discoid cells up to 25 μm wide, and enveloped in a distinct sheath. Filament morphology, structurally preserved by permineralization and mineral replacement, corresponds to the fossil genus *Palaeolyngbya*, a predominantly Precambrian equivalent of the extant *Lyngbya* sensu lato (Oscillatoriaceae, Oscillatoriales). Specimens occur exclusively in masses of interwoven hyphae produced by the fungus *Endochaetophora antarctica*, suggesting that a special micro-environmental setting was required to preserve the filaments. Whether some form of symbiotic relationship existed between the fungus and cyanobacterium remains unknown.

## INTRODUCTION

Cyanobacteria, one of the most successful groups of prokaryotic microorganisms on Earth, were instrumental in the oxygenation of the atmosphere and, as primary producers and nitrogen-fixers, were and are prominent contributors to Earth's nutrient cycles (*Knoll, 2008*). The fossil record of these life forms is extensive and varied, with a peak of documented morphologies and formally described taxa in the Proterozoic (∼2.54–0.54 Ga) (*Tomitani et al., 2006*; *Knoll, 2008*; *Schopf, 2012*).

Fossil cyanobacterial filaments that correspond in morphology to the modern genus *Lyngbya* (*Komárek et al., 2014*; *Guiry & Guiry, 2019*) are ordinarily assigned to *Palaeolyngbya*, a fossil taxon for wide, unbranched filaments composed of cylindrical trichomes with discoid cells several times wider than long, and colorless sheaths

Corresponding author
Carla J. Harper, harper@snsb.de, charper@ku.edu

(e.g., *Schopf, 1968*). *Palaeolyngbya* is primarily Proterozoic and Cambrian in age (see references in *Butterfield, Knoll & Swett, 1994*; *Sergeev, Sharma & Shukla, 2012*). However, exquisitely preserved specimens have been reported recently from the Lower Devonian Rhynie chert (*Krings, 2019*), and there is also one record from the Permian of China (*Liu & Li, 1986*: pl. 1, fig 5). The genus has not yet been documented from the Mesozoic, whereas Cenozoic fossils and subfossil specimens are commonly assigned to *Lyngbya* (e.g., *Waggoner, 1994*; *Stankevica et al., 2015*).

Permineralized peat from Fremouw Peak in the central Transantarctic Mountains, Antarctica, represents a unique source of new information on Middle Triassic (240 Ma) high-latitude swamp-forest ecosystems. The peat contains an exceptionally diverse structurally preserved flora (reviewed by *Escapa et al., 2011*; *Bomfleur et al., 2013*; *Bomfleur et al., 2014*; *Decombeix et al., 2014*), together with numerous examples of fungi and fungus-like organisms (reviewed by *Harper et al., 2016*). However, no evidence of the occurrence of photoautotrophic microorganisms, such as cyanobacteria and eukaryotic algae, in the Fremouw Peak permineralized peat has been discovered to date.

This paper presents the first record of a filamentous cyanobacterium from the Fremouw Peak permineralized peat. Specimens are similar morphologically to *Palaeolyngbya kerpii* from the Lower Devonian Rhynie chert (*Krings, 2019*), and to several Proterozoic species of that genus (*Schopf, 1968*; *Butterfield, Knoll & Swett, 1994*; *Sergeev, Sharma & Shukla, 2012*). The Antarctic filaments all occur within masses of interwoven hyphae produced by a fungus. This discovery is important because it provides insights into the taphonomic circumstances that appear to be imperative to the preservation of cyanobacteria in permineralized peat.

## MATERIAL & METHODS

Data were collected from the same locality previously described by *Harper et al. (2015)*, specifically the fossils occur in permineralized (silicified) peat from the Fremouw Formation in the central Transantarctic Mountains of Antarctica (*Taylor, Taylor & Collinson, 1986*; *Cúneo et al., 2003*). The Fremouw Formation is a 620–750-m-thick siliclastic succession deposited by low - sinuosity braided streams (*Faure & Mensing, 2010*). The fossils occur within several allochthonous clasts that are at approximately the same stratigraphic level within a trough cross-bedded, medium-grained, greenish-gray volcaniclastic sandstone. Permineralized peat is found at a single level at the Fremouw Peak locality, approximately 30 m below the top of the formation (Fig. 1). Chunks of the peat were likely rafted into their current position during a flooding event that caused them to be stranded on sand bars prior to permineralization (*Taylor, Taylor & Collinson, 1989*) and isolated into individual lenses within the outcrop. The peat became silicified after burial; the age of the plant remains contained in the peat is equivalent to that of the surrounding clastic sediments, i.e., fluvial sandstone, which also contain trunks of wood of equivalent age to the peat (*Decombeix et al., 2014*). The silica for the permineralization is interpreted to have come from the dissolution of abundant siliceous, volcanic detritus from the upper Fremouw Formation (*Taylor, Taylor & Collinson, 1989*).

The exact age of the Fremouw Peak peat deposit remains uncertain. The peat and surrounding material have been dated as Anisian (early Middle Triassic) based on

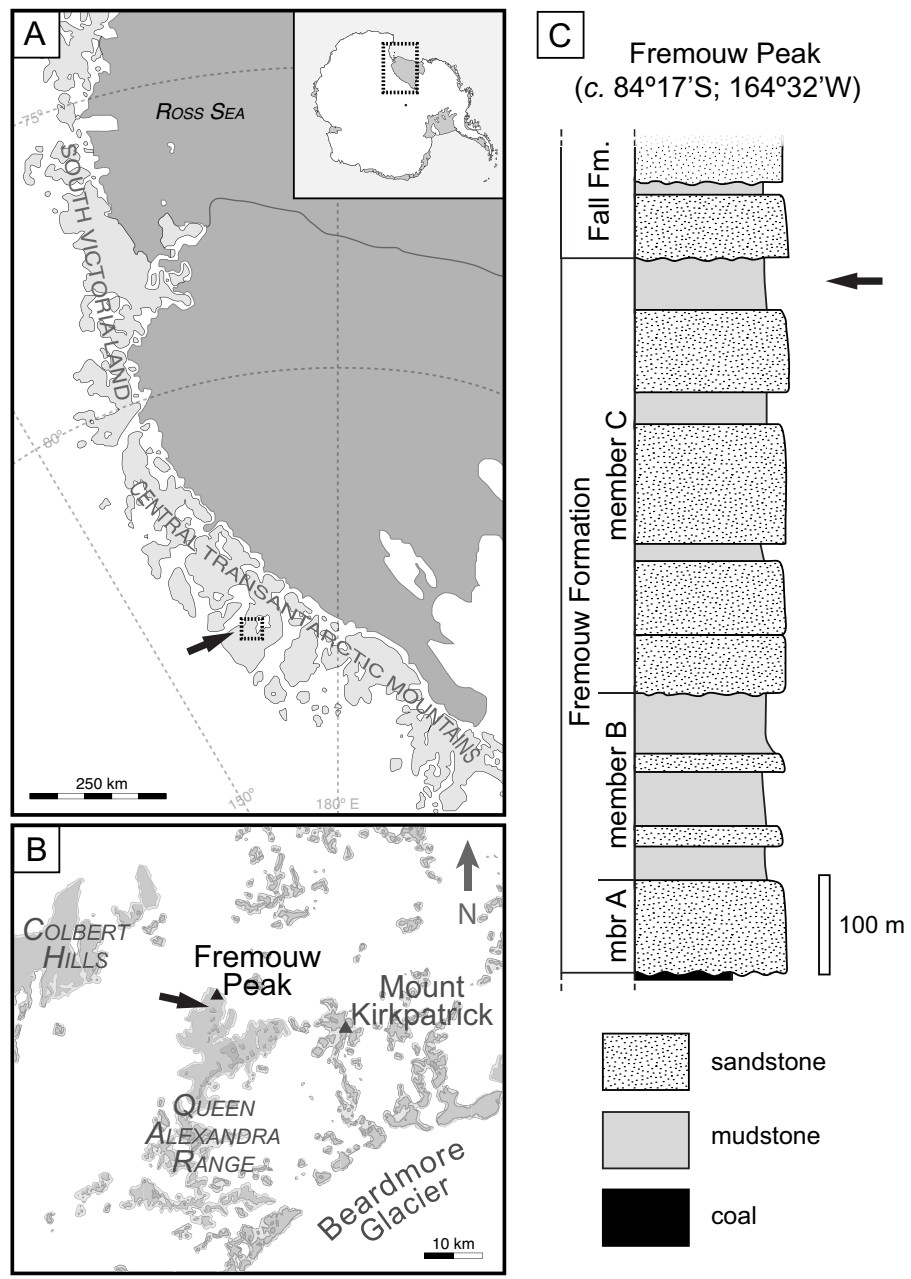

**Figure 1** Geographic occurrence and stratigraphic position of the Fremouw Peak permineralized peat; modified from **Fig. 1** in "Habit and ecology of the petriellales, an unusual group of seed plants from the Triassic of Gondwana" by *Bomfleur et al. (2014)*©2014 by The University of Chicago. All rights reserved. (A) Overview of collection area in the Central Transantarctic Mountains, South Victoria Land, Antarctica. (B) Boxed area and arrow in (A). Detail of Fremouw Peak locality with arrow indicating collecting site. (C) Stratigraphic column of Fremouw Peak locality with arrow indicating position of permineralized peat.

palynomorphs and nearby vertebrate fossils (*Farabee, Taylor & Taylor, 1990*; *Hammer, 1990*; *Sidor, Damiani & Hammer, 2008*; *Faure & Mensing, 2010*). Recent detrital-zircon dating indicates that the base of member B of the Fremouw Formation is ∼242.3 ± 2.3

Ma (=early Ladinian; see *Elliot et al., 2017*: fig. 4), but the silicified peat is located in the younger member C of the Fremouw Formation. A late Ladinian or possibly Carnian age is, therefore, more likely to be accurate for the Fremouw Peak peat deposits (*Bomfleur et al., 2014*; *Elliot et al., 2017*).

The material used in this study was collected during the 2010–2011 austral summer Antarctic field season. Peat blocks were cut into slabs and then immersed in 48% HF to dissolve the silica. Acetate peels were produced from the etched surfaces by using the technique outlined by *Joy, Willis & Lacey (1956)* modified for hydrofluoric acid (*Galtier & Phillips, 1999*). Consecutive peels of promising specimens were mounted on microscope slides in Eukitt®. Other slabs were cut into wafers and used for the preparation of thin sections (*Hass & Rowe, 1999*), with a thickness of 40–60 µm. Wafers of the peat were cemented to a glass slide and then ground thin enough to be viewed in transmitted light. Mounted peels and thin sections were analyzed with a Leica DM LB2 transmitted light microscope at the highest possible total magnification (400× or 1,000×); digital images were captured with a Leica DFC-480 camera and processed in Adobe Photoshop CS5. When suitable specimens were identified, they were processed minimally (i.e., contrast, brightness, and focal stacking) and measurements were taken using Adobe Photoshop CS6 Version 13.0 × 64 (Adobe Systems, San Jose, CA, USA). When necessary, multiple images of the same specimen were recorded at different focal planes and compiled to produce composite images, (*Kerp & Bomfleur, 2011*). The images were stacked in Adobe Photoshop CS6, and specific areas were modified to reveal the complete three-dimensional image as seen in the thin section. Composite images in this study are Figs. 2A–2C. Specimen and slides are deposited in the Paleobotanical Collections, Biodiversity Institute, University of Kansas (KUPB) under specimen accession numbers KUPB 17054, 17729 E Bot, 17729 F Top, and 18084, and slide numbers KUPB 35,009–35,018.

## RESULTS

### Context and preservation

Systematic screening of permineralized peat from the Fremouw Peak locality has yielded several hundred blocks of leaf mats that contain predominantly degraded *Dicroidium* leaves (*Pigg, 1990*) ("L" in Figs. 2A and 2B), rare pieces of fragmented *Heidiphyllum* (*Axsmith, Taylor & Taylor, 1998*), degraded plant axes, and intermixed detritus. Some of the leaves are surrounded by conspicuous whitish areas, which are elongate oval or irregular in section view, 120–590 µm high, and up to 4 cm wide. The whitish areas comprise densely interwoven, thin-walled, irregularly septate fungal hyphae 2–6 µm wide (arrows in Fig. 2C) embedded in what appears to be a gelatinous matrix. More than 95% of these formations, henceforth called "hyphal masses", contain one or several specimens of the enigmatic fungal reproductive unit *Endochaetophora antarctica* (Figs. 2A and 2B), formally described some 30 years ago based on dispersed specimens from the same locality (*White & Taylor, 1988*; *White & Taylor, 1989*). For a parallel study focusing on *E. antarctica,* we analyzed more than 50 blocks of leaf mats based on thin sections, each containing between 1 and 15 hyphal masses. In these blocks, approximately 25% of the larger masses containing

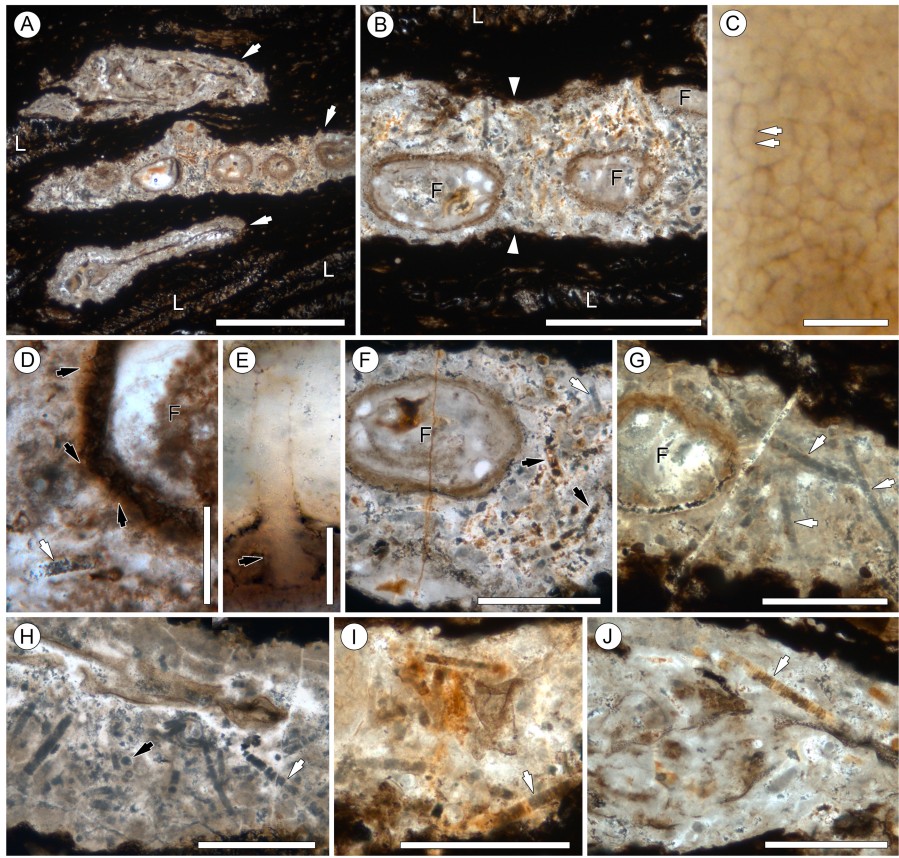

**Figure 2 Overview of *Endochaetophora antarctica* hyphal masses and *Palaeolyngbya* sp. in permineralized peat.** (A) Three hyphal masses (arrows) in leaf mats (L); slide KUPB 35,009; scale bar = 1 cm. (B) Higher magnification of Fig. 2A, showing hyphal mass (between arrowheads) and *E. antarctica* reproductive units (F); slide KUPB 35,009; scale bar = 500 μm. (C) High magnification of densely spaced hyphae comprising hyphal mass; arrows indicate septa; slide KUPB 35,017; scale bar = 10 μm. (D) Comparison of appendages (black arrows) of *E. antarctica* fungal reproductive unit (F) to adjacent *Palaeolyngbya* filament (white arrow); slide KUPB 35,017; scale bar = 50 μm. (E) High magnification of *E. antarctica* appendage; portion of appendage extending into hyphal mass and base of appendage in wall of *E. antarctica* (arrow); slide KUPB 35,018; scale bar = 10 μm. (F) Hyphal mass containing *E. antarctica* (F) and fragmented cyanobacterial filaments; note different mineral replacement, reddish-orange filaments (black arrows) and gray mineral (white arrow); slide KUPB 35,009; scale bar = 250 μm. (G) Hyphal mass with *E. antarctica* (F) and long cyanobacterial filaments (arrows); slide KUPB 35,010; scale bar = 250 μm. (H) Assemblage of cyanobacterial filaments in hyphal mass; filaments in cross (black arrow) and longitudinal section views (white arrow); slide KUPB 35,010; scale bar = 250 μm. (I) Assemblage of cyanobacterial filaments preserved as reddish-orange mineral replacements; note detail of discoid cells (arrow); slide KUPB 35,011; scale bar = 500 μm. (J) Well preserved filament in hyphal mass (arrow); slide KUPB 35,009; scale bar = 500 μm.

fungal reproductive units also contain large cyanobacterial filaments, which are described below. *Endochaetophora antarctica* is characterized by a three-layered investment from which extend numerous prominent hollow, tube-like appendages (∼4.5–10 μm wide and up to 130 μm long) that branch regularly (Figs. 2D and 2E). Because the appendages

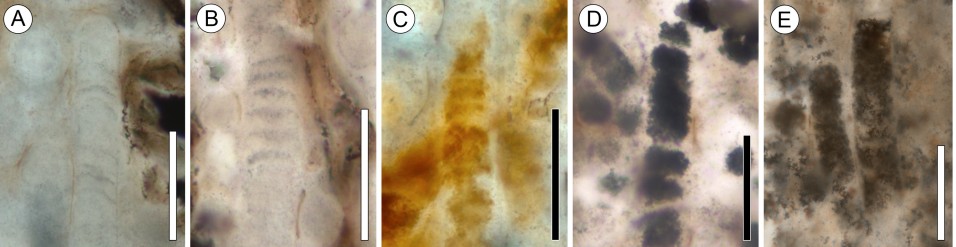

**Figure 3  Range of preservation states of cyanobacterial filaments.** All scale bars = 50 μm. (A) Clear mineral replacement with barely visible discoid cells. Note rounded (possible) filament tip; slide KUPB 35,009. (B) Clear mineral replacement with discoid cells well recognizable; slide KUPB 35,009. (C) Reddish-orange mineral replacement; slide KUPB 35,011. (D) Cell lumina filled with opaque matter; slide KUPB 35,010. (E) Filaments with fine granular opaque matter; slide KUPB 35,010.

are markedly different structurally from the cyanobacterial filaments, the two structures cannot be confused. The cyanobacterial filaments are not body fossils as the hyphal masses containing the fungal reproductive units, but rather represent (partial to full) mineral infillings or coatings, which are orange to reddish or have grayish outlines (Figs. 2F–2J; 3A–3E). Cyanobacterial filaments have not been found in the peat matrix surrounding the hyphal masses, or elsewhere in the peat.

## Cyanobacterial filaments

In the description of the fossil cyanobacterium, we use the terminology for filamentous cyanobacteria outlined by *Komárek, Kling & Komárková (2003)*; trichomes with sheaths are traditionally termed filaments. Preserved filament portions (arrows in Figs. 2G–2J) are up to 740 μm long and 17–31 μm wide, and consist of straight or somewhat curved, cylindrical, uniseriate, and probably isopolar (i.e., no evidence indicative of heteropolarity has been found) trichomes of relatively uniform, short discoid cells, enveloped in a distinct sheath (Figs. 4A–4C). Most of the specimens demonstrate a regular pattern of discoid cells (or cell units), which are either empty or contain homogenous opaque matter (cell contents), 22.8–25 μm wide and 3.8–5 μm high (which equals a width-to-height ratio of 5:1) (Fig. 4A). Other specimens, however, are preserved as empty sheaths (Fig. 4F), whereas in still others the cells are recognizable only through the arrangement of crystals (red to orange, or gray in appearance) (Figs. 2G, 2A–2E, 4E). Different modes of cell and trichome preservation may also occur within the same filament (Fig. 2F).

Sheaths are colorless, unornamented, and well-recognizable in all specimens. They range in thickness from 1 to 4 μm; however, sheath thickness within one filament varies only by 1–1.5 μm. Stratification of the sheath is not recognizable in any of the specimens; external constrictions or folds at cross walls are also not discernible. Most specimens represent intercalary trichome portions that end bluntly and appear to have broken off. Compelling evidence of trichome tips has not been found. There is a single poorly preserved specimen that appears to have a tapering tip with a round end; however, it is difficult to be sure that this represents an actual trichome end (Fig. 3A, black arrow in 2G). Unfortunately, the preservation of the filaments by mineral replacement does not enable recognition of

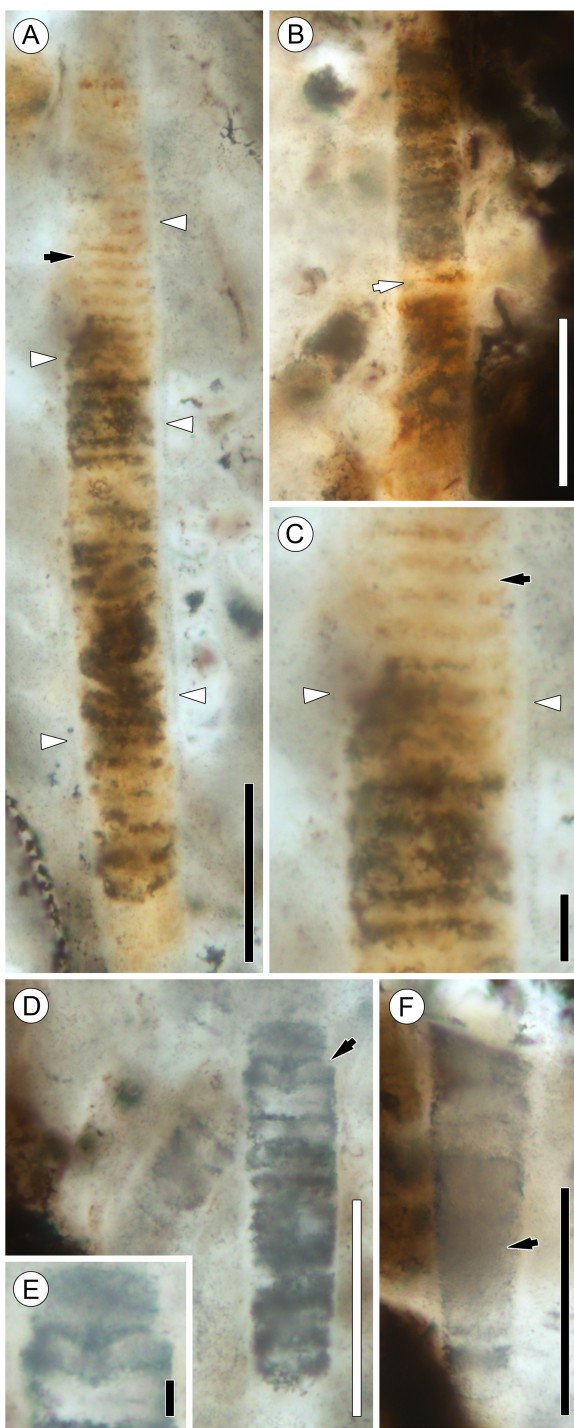

**Figure 4 Details of *Palaeolyngbya* sp. filaments.** (A) Overview of trichome with prominent sheath (arrowheads) and discoid cells (arrow); slide KUPB 35,009; scale bar = 50 μm. (B) Filament portion with possible necridium (arrow); slide KUPB 35,009; scale bar = 50 μm. (C) High magnification of Fig. 4A, showing filament with discoid cells (black arrow) and prominent sheath (arrowheads); slide KUPB 35,009; scale bar = 10 μm. (D) Filament with constriction (arrow); slide KUPB 35,012; scale bar = 50 μm. (E) High magnification of constriction in Fig. 4D; slide KUPB 35,012; scale bar = 5 μm. (F) Filament showing portion of trichome in which cells are not preserved (arrow); slide KUPB 35,012; scale bar = 50 μm.

cell division patterns. One specimen shows possible hormogonium formation (Figs. 4D and 4E). This filament is the only example displaying a pronounced constriction, and the cell at the constriction is umbilicated. One intercalary filament portion approximately 87 µm long might contain a necridium based on the presence of a pair of differently shaped and colored cells (Fig. 4B). No evidence has been found of (false) branching and the formation of heterocysts or akinetes. For a graphical overview of the spatial distribution of cyanobacterial filaments and *E. antarctica* within one of the hyphal masses, refer to Fig. 5.

## DISCUSSION

The Triassic permineralized peat from Fremouw Peak has been studied intensively for more than 45 years (e.g., *Schopf, 1973*; *Taylor, Taylor & Collinson, 1989*). Plant and fungal paleodiversity have been documented in great detail based on large numbers of structurally preserved fossils (e.g., *Escapa et al., 2011*; *Harper et al., 2016*), and the paleoecosystem has been reconstructed as a diverse peat-forming swamp forest dominated by corystospermalean seed ferns and voltzialean conifers, with understory elements including the enigmatic Petriellales, ferns, and sphenophytes (*Taylor, Taylor & Collinson, 1989*; *Escapa et al., 2011*; *Bomfleur et al., 2014*; *Decombeix et al., 2014*). However, there is not a single report of cyanobacteria from Fremouw Peak despite these organisms being regular constituents of comparable modern peat-forming ecosystems (*Jackson, Liew & Yule, 2009*; *Yule & Gomez, 2009*; *Marsid et al., 2015*). Fossils of cyanobacteria have been described from Antarctica (*Priestley & David, 1912*; *David & Priestley, 1914*; *Chapman, 1916*; *Gordon, 1921*; *Hill, 1964*; *Breed, 1971*; *Rees, Pratt & Rowell, 1989*; *Rowell & Rees, 1989*; *Riding, 1991*; *Wrona & Zhuravlev, 1996*; *Wrona, 2004*); however, none come from the Mesozoic.

### Comparison and affinities

The Fremouw Peak cyanobacterial filaments correspond in morphology to the fossil genus *Palaeolyngbya*, a form taxon and repository for wide, unbranched, uniseriate fossil trichomes that are composed of discoidal to cylindrical cells without any constrictions at the cross walls, and enveloped in a prominent, uni- or multilayered, smooth sheath, and hence comparable in basic organization to extant *Lyngbya* sensu lato (Oscillatoriaceae, Oscillatoriales see *Butterfield, Knoll & Swett, 1994*: p. 60/61; *Sergeev, Sharma & Shukla, 2012*: p. 300). The main criterion used to discriminate species of *Palaeolyngbya*, according to *Butterfield, Knoll & Swett* (*1994*: p. 61), is the width of the uncollapsed sheath (i.e., filament width); for example, (sheath width in parentheses) *P. catenata* (10–30 µm) and *P. hebeiensis* (30–60 µm) (*Butterfield, Knoll & Swett, 1994*), *P. giganteus* (42–85 µm), *P. helva* (11–14 µm), and *P. barghoorniana* (≤15 µm) (*Sergeev, Sharma & Shukla, 2012*), and *P. kerpii* (22–>30 µm) (*Krings, 2019*). The sheaths of the Fremouw Peak filaments are 17–31 µm wide and, thus, correspond best to the recently described *P. kerpii* from the Lower Devonian Rhynie chert. Assignment of the Triassic filaments to *P. kerpii* is conceivable. However, *P. kerpii* is exquisitely preserved as a petrification providing detailed insights into filament morphology, together with specific developmental details, whereas the Fremouw Peak fossils represent mineral replacements that provide a fair image of the filament

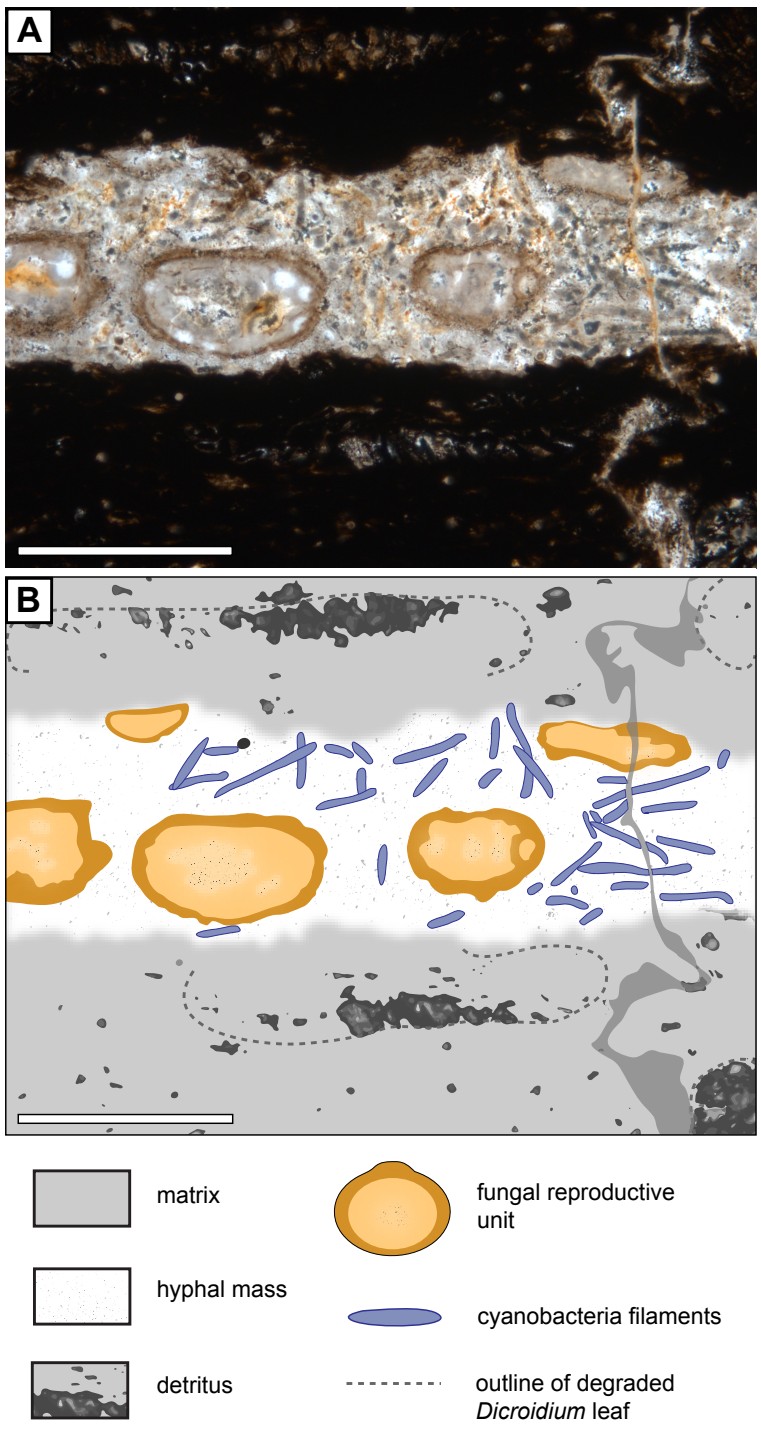

**Figure 5 Association of *Endochaetophora antarctica* with *Palaeolyngbya* sp. in permineralized peat.**
(A) Photograph of *Endochaetophora antarctica* and *Palaeolyngbya* sp. in permineralized peat. (B) Graphical representation of Fig. 5A, showing distribution of cyanobacterial filaments in *Endochaetophora antarctica* hyphal mass. Scale bars = 500 µm.

morphology, but do not reveal any structural or developmental details. As a result, it is difficult, if not impossible, to determine whether the latter correspond to *P. kerpii* or belong to a different fossil species. Therefore, we include the Fremouw Peak filaments in open nomenclature as *Palaeolyngbya* sp.

## Cyanobacteria in Triassic permineralized peat

One reason for the lack hitherto of documented evidence of cyanobacteria in the Triassic permineralized peat from Fremouw Peak may be that these minute life forms simply have been overlooked in cursory screenings of peels or thin sections at low magnification. Moreover, the quality of plant fossils preserved in the peat matrix depends largely on their condition (i.e., alive and fully intact, moribund but still attached, or abscised and in the process of degradation) at the time of permineralization. Evidence of microbial life appears to be generally rare in regions of the peat that contain well-preserved plant remains, but rather occurs in peat comprising (partially) degraded and tattered plant material not worthwhile for investigators interested in the plants and, thus, are often not seen (*Taylor & Krings, 2010*). On the other hand, the Fremouw Peak permineralized peat is interpreted to have developed in a three-step process (*Schopf, 1971*; *Taylor, Taylor & Collinson, 1989*), through which fragile structures may have been altered secondarily or destroyed (*Harper et al., 2018*). Finally, the lack of evidence for these organisms from permineralized peat elsewhere, (e.g., *DiMichele & Phillips, 1994*; *Galtier, 2008*; *McLoughlin & Strullu-Derrien, 2015*; *Slater, McLoughlin & Hilton, 2015*), could mean that peat-forming paleoenvironments were perhaps generally not conducive to the preservation of cyanobacteria. The scarcity of cyanobacterial fossils in peat deposits stands in stark contrast to silicified geothermal hot spring (sinter) deposits, which often yield diverse assemblages of structurally preserved cyanobacteria (e.g., *Guido et al., 2010*; *García Massini et al., 2012*; *Hamilton et al., 2019*; *Krings, 2019*; *Krings & Harper, 2019*; *Krings & Sergeev, 2019*). Nothing is known to date about the possible influence of a hydrothermal system on the silicification process at Fremouw Peak (*Taylor, Taylor & Collinson, 1989*).

Cyanobacterial filaments have only been detected in the whitish hyphal masses produced by *Endochaetophora antarctica* around individual *Dicroidium* leaves on the forest floor (*Harper, 2015*). Because the filaments are salient structures, we rule out the possibility that they have been overlooked in the peat matrix surrounding the hyphal masses and in other types of fossils, such as hollow plant axes or decayed leaves. This peculiar pattern of spatial distribution raises the question as to why filaments are so abundant in the *E. antarctica* hyphal masses, but are absent (or cannot be traced) outside these occurrences?

One possible explanation is that a special micro-environmental setting was imperative for the filaments to become preserved intact. Research on fragile microorganisms, including cyanobacteria, exquisitely preserved elsewhere has provided evidence to suggest that certain micro-environmental settings (e.g., amber, walls of leech cocoons, interiors of hollow plant axes, small voids in the substrate, or microbial mat frameworks) had a cushioning effect on destructive mechanical forces, and hence were effective as microscopic conservation traps for delicate microbial life (*Dörfelt, Schmidt & Wunderlich, 2000*; *Bomfleur et al., 2012*; *Bomfleur et al., 2015*; *McLoughlin et al., 2016*; *Krings et al., 2018*; *Krings & Harper, 2019*;

*Krings & Kerp, 2019*). It is highly probable that special circumstances also were in play during the fossilization of the cyanobacteria from Fremouw Peak. The hyphal masses, which are embedded in what appears to be a gelatinous matrix, may have served as a conservation trap by shielding the filaments from destructive mechanical forces, such as water movement and the taphonomic processes during peat formation, compaction, and permineralization. Moreover, certain substances excreted by the fungal hyphae into the surrounding matrix may have been biocidal and slowed down biological decomposition, or somehow facilitated the process of mineral replacement. If all this is accurate, then it raises another, equally difficult and probably even more complex question, namely as to why cyanobacterial filaments occur in large numbers within hyphal masses produced by a fungus.

Cyanobacteria in general (*Dickinson, 1983*; *Zadorina et al., 2009*; *Andersen, Chapman & Artz, 2013*), and certain members of *Lyngbya* in particular (*Karosiene & Kasperovičiene, 2009*; *Koreiviene, Kasperovičiene & Karosiene, 2009*), are constituents of modern peatland environments, and it would not be surprising to find these organisms also in ancient peat-forming ecosystems based on their geologic range. However, the opposite is the case. *Palaeolyngbya* filaments (and other cyanobacteria) were perhaps common and widespread on the wet forest floor covered in leaf litter interspersed with *E. antarctica* hyphal masses, in small pools of water, and maybe even on tree surfaces, but were destroyed during peat formation and the fossilization process, with the exception of those located within the protective confines of the hyphal masses ("cyanobacteria everywhere hypothesis"; see Fig. 6). On the other hand, metagenomic analyses indicate that cyanobacteria represent a relatively small percentage of total microbial biomass in modern peat ecosystems, namely 0–4% in peat bogs and ∼0.85% in tropical peat swamps (*Gilbert & Mitchell, 2006*; *Kanokratana et al., 2011*). Thus, it is also possible that *Palaeolyngbya* and other cyanobacteria have not been recorded more extensively because they were rare elements in this type of paleoenvironment or occurred exclusively in certain areas of the ecosystem that are not reflected by the silicified peat samples studied to date (see *Krings & Sergeev, 2019*).

Although the systematic affinity of *Endochaetophora antarctica* remains unresolved, several authors have suggested it may belong to the Mucoromycota (see discussion by *Krings, Taylor & Dotzler, 2013*). Specimens of another fossil fungus from Fremouw Peak, *Jimwhitea circumtecta*, provide the most persuasive fossil example of spores forming within a sporocarp and embedded in what is commonly termed a gleba (*Krings et al., 2012*: figs. 2B–2C). The hyphal masses of *E. antarctica* are certainly not glebae in the strict sense of the definition (i.e., the central, internal portion of a fruiting body see *Ulloa & Hanlin, 2012*: p. 252), but may be analogous structures within which the reproductive units formed. The chemical composition of glebae in Mucoromycota is virtually unknown; however, glebae of certain present-day Basidiomycota are composed primarily of amino acids and proteins (*Oliveira & Morato, 2000*). It is, therefore, a possible, although highly speculative alternative premise at this time, that the cyanobacteria were attracted to the components of the *E. antarctica* hyphal masses and, therefore, migrated into these structures ("cyanobacteria migration hypothesis"; see Fig. 7). Bearing in mind that the

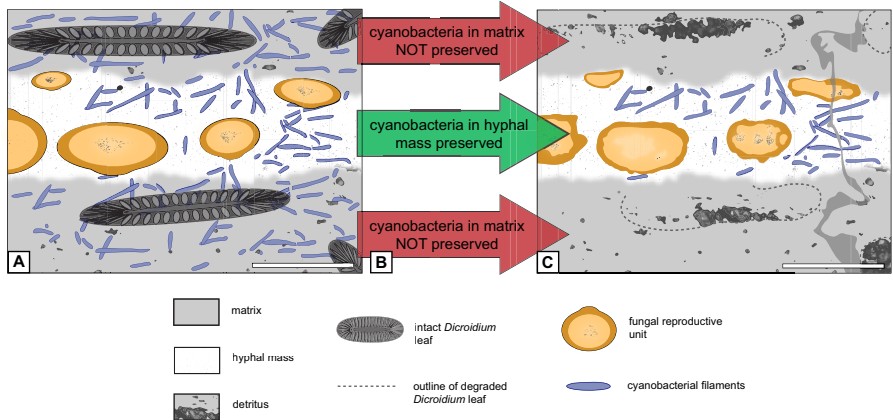

matrix

hyphal mass

detritus

intact *Dicroidium* leaf

outline of degraded *Dicroidium* leaf

fungal reproductive unit

cyanobacterial filaments

**Figure 6** Graphical representation of "cyanobacteria everywhere" hypothesis. (A) Filaments occur everywhere in matrix and hyphal mass. (B) Filaments not preserved in peat but in hyphal mass. (C) Filaments found exclusively in hyphal mass. Scale bars = 500 µm.

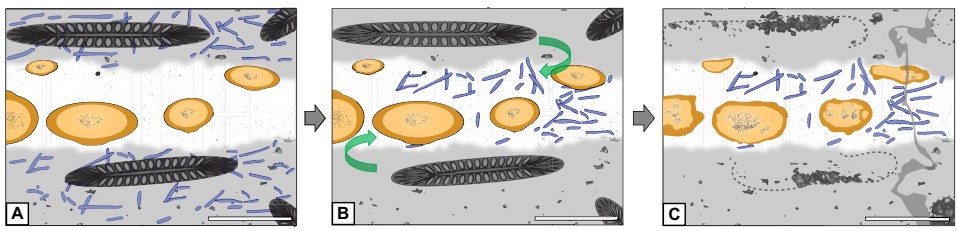

**Figure 7** Graphical representation of the "cyanobacteria migration" hypothesis. (A) Filaments occur exclusively in matrix. (B) Filaments migrate into hyphal mass. (C) Filaments found exclusively in hyphal mass. Refer to key in Fig. 6. Scale bars = 500 µm.

water in peat-forming environments today is generally nutrient-poor and of low pH, it is possible to envision that the cyanobacteria would gravitate towards nutritionally dense resources. Moreover, it has been shown that, under certain stimuli or in high stress environments, some present-day filamentous cyanobacteria actively migrate towards and assimilate specific amino acids (*Gallucci & Paerl, 1983*; *Michelou, Cottrell & Kirchman, 2007*).

## A symbiosis?

No direct evidence has been found to date that is suggestive of an interaction between the *Palaeolyngbya* filaments and *Endochaetophora antarctica*, nor has the nutritional mode of *E. antarctica* been deciphered. Nevertheless, the consistent co-occurrence of these two organisms begs the question as to whether this peculiar alliance may also have included some form of mutualism or parasitism. Mutualistic relationships between filamentous cyanobacteria and fungi today occur in the form of lichens (*Hawksworth, 1988*). The Fremouw Peak fossils concur with some of the criteria outlined by *Lücking & Nelsen* (*2018*: p. 552) for the identification of fossil lichens; the most important criterion, however, namely

a physiological interdependence between the partners, cannot be evidenced. Another type of fungal symbiosis with filamentous cyanobacteria occurs in *Geosiphon pyriformis*, a fungus in the Glomeromycota that produces specialized bladders to harbor nitrogen-fixing cyanobacteria (*Nostoc* spp.) (*Schüßler, 2002*; *Schüßler, 2012*). *Palaeolyngbya* is non-heterocystous; however, certain non-heterocystous filamentous cyanobacteria, including *Lyngbya* under extremely stressful conditions, can also fix nitrogen (*Bergman et al., 1997*). In addition, some authors include *Geosiphon* within Mucoromycota (Glomeromycotina and Mucoromycotina) (*Spatafora et al., 2016*), to which also *E. antarctica* probably belongs. We speculate that perhaps there were extinct members of the Mucoromycota that formed non-lichen symbioses with cyanobacteria, and that the *Geosiphon Nostoc* symbiosis represents a relic of this type of fungus-cyanobacterial symbiosis (*Schüßler, 2002*), which not only involved fungi interacting with endocytobiotic cyanobacteria, but perhaps also forms that housed their cyanobacterial symbionts in hyphal masses. On the other hand, there is also the remote possibility that the fungus parasitized the cyanobacteria, which were somehow attracted into the hyphal masses (e.g., *Arora, Filonow & Lockwood, 1983*).

## CONCLUSIONS

*Palaeolyngbya* in the Triassic permineralized peat from Fremouw Peak provides the first evidence of filamentous cyanobacteria from the Mesozoic of Antarctica. Moreover, the restricted occurrence of the cyanobacterial filaments within hyphal masses produced by a fungus suggests that special micro-environmental conditions have preserved these organisms in recognizable form, and that the fungal hyphal masses have served as microscopic conservation traps for microbial life (*sensu Bomfleur et al., 2012*). The recognition of cyanobacteria in microscopic conservation traps provides a search image that now can be used to trace this and other types of microorganisms in the vast amounts of permineralized peat that have been collected from Fremouw Peak. We anticipate that other cyanobacteria will be discovered as further special micro-environmental settings conducive to the preservation of microbial life are identified. The information obtained from studying the microbial component of Antarctic Mesozoic paleoecosystems may help to address questions pertaining specifically to the ecology of high–latitude plants and paleoecosystems, including such aspects of whether the only fossil cycad that has been documented to date from Antarctica, *Antarcticycas schopfii* (*Hermsen, Taylor & Taylor, 2009*), entered into a symbiotic relationship with cyanobacteria in a similar way as its relatives today (e.g., *Lindblad & Bergman, 1990*; *Costa & Lindblad, 2002*; *Tajhuddin et al., 2010*).

## ACKNOWLEDGEMENTS

We thank R. Serbet (Lawrence, KS, USA) for technical assistance, Benjamin Bomfleur (Münster, Germany) for permission to use and modification of Fig. 1, D. Elliot (Columbus, OH, USA) for assistance with mineralogy and Antarctic geology, A.-L. Decombeix (Montpellier, France) for fruitful discussion, P.A. Penhale (OPP NSF, Alexandria, VA) for assistance with questions on collection permits, and the University of Kansas Interlibrary

Loan department for help in procuring obscure literature. We thank our handling editor, C. Moyer (Bellingham, WA, USA), and two anonymous referees for their constructive comments and suggestions.

### Funding

Funding for this work was provided by the United States National Science Foundation (U.S. NSF-1443546 to Edith L. Taylor). The funders had no role in study design, data collection and analysis, decision to publish, or preparation of the manuscript.

### Grant Disclosures

The following grant information was disclosed by the authors:
United States National Science Foundation: U.S. NSF-1443546.

### Competing Interests

The authors declare there are no competing interests.

### Author Contributions

- Carla J. Harper conceived and designed the experiments, performed the experiments, analyzed the data, prepared figures and/or tables, authored or reviewed drafts of the paper, and approved the final draft.
- Edith L. Taylor conceived and designed the experiments, analyzed the data, authored or reviewed drafts of the paper, provided resources to conduct research, and approved the final draft.
- Michael Krings conceived and designed the experiments, performed the experiments, analyzed the data, authored or reviewed drafts of the paper, provided resources to conduct research, and approved the final draft.

### Data Availability

Specimen and slides are deposited in the Paleobotanical Collections, Biodiversity Institute, University of Kansas (KUPB), Lawrence, Kansas, USA: KUPB 17054, 17729 E Bot, 17729 F Top, and 18084, and slide numbers KUPB 35,009–35,018.

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
