# Peer review of "Filamentous cyanobacteria preserved in masses of fungal hyphae from the Triassic of Antarctica"

_PeerJ, doi:10.7717/peerj.8660_

## Round 0.1 · original submission · Minor Revisions

I agree with the reviewers in that for the most part, this is a well-organized and well-written manuscript. There are a few points that require the authors attention, that will undoubtedly improve the outcome.

Reviewer 1 ·

Basic reporting

The manuscript is basically well written and structured. See specific comments below and on the attached pdf.

Experimental design

This manuscript is within the scope of the journal. It provides novel insights into the microbial communities of terrestrial peat forming ecosystems of Antarctica that have taught us much about Triassic plant anatomy and forest structure in recent decades.

Validity of the findings

The conclusions are valid and the material has not been "over interpreted" as might have been tempting for such a curious association of cyanobacteria and fungi within matted leaf laminae. Three hypotheses are presented to account for the fungal - cyanobacterial association.

Additional comments

This manuscript documents the first Mesozoic cyanobacterial fossils from Antarctica. They are uniquely preserved in fungal hyphal masses within a leaf-rich permineralized (silicified) peat.
The new record makes a significant addition to the biotic content and palaeoecology of Antarctic permineralized peats, which have yielded extensive data for the reconstruction of vascular plants and fundi over several decades. The documented association between the fungus and the cyanobacterium is intriguing. The authors raise three hypotheses to explain the fungal-cyanobacterial association ranging from chance co-occurrence in the peat matrix to an obligate lichen-like association. Although, they do not find conclusive evidence to support any one or other of these hypotheses, the strong apparent linkage between the two organisms suggests some mutually beneficial relationship.
Specific comments:
1. The paper is well written. I have marked up a few items on the attached pdf as suggestions for improving the punctuation and grammar.
2. Somewhere early in the text, the authors should clarify that the term trichome refers to the central column of cells and that the term filament refers to the trichome and its surrounding sheath. This may be obvious to specialists on cyanobacteria but will not be obvious to most. In a few cases through the manuscript, the terms trichome and filament almost appear to be used interchangeably.
3. The age of the permineralized peat given in the introduction (240 Ma = late Middle Triassic) is slightly at odds with that given in the Materials and methods (Middle or Late Triassic). This just needs some simple rephrasing of the Introduction.
4. The references are in good order. I have listed a couple of additional references on the attached pdf that might be relevant to this study.
5. Cyanobacteria are rare or virtually unknown from other woody (forested) permineralized peats. However, they are quite common in ancient sinter deposits and commonly form matted fabrics – see references mentioned on the attached pdf and other papers by Malcolm Walter and Jack Farmer. Could this provide some evidence for the Fremouw Peak peats being influenced by ancient thermal spring systems?
6. The illustrations are good considering the limitations of the preservational quality of the material. The interpretative drawings are a useful complement to the study. I think Fig. 2C needs some additional explanation. The reticulate patterning is distinctive and I have seen something similar to this feature before in fossil wood. Does this feature represent a reticulate mass of hyphae (in which case each hypha would be very small at a few microns width) or does it represent a single hypha with a reticulate wall ornamentation (in which case the hypha would be quite large at around 25-30 microns width).

Annotated reviews are not available for download in order to protect the identity of reviewers who chose to remain anonymous.

Reviewer 2 ·

Basic reporting

The language used throughout the manuscript is generally clear and unambiguous. The fossil descriptions are as thorough as the preservation allows and the authors make it clear when they are speculating on the features or ecological roles of the cyanobacterium. The discovery of this fossil is placed in an appropriate context and shows a sufficient familiarity with the literature. The figures are clear and generally well-labeled and described within the text and figure captions. As this is a descriptive paper, there was no raw data to view other than the images presented. This is a self-contained manuscript and does not appear to have been cut from a larger related study.

With that being said, I do have a few suggestions for improvements, clarifications, removal of extraneous words, and fixes for typos/errors.

Line 168: The authors reference a black arrow in Figure 4G, but Figure 4G does not exist in this manuscript. Update this to reference the appropriate figure.

Figure 1: Although the it is fairly self-evident that the arrows indicate the location of the Fremouw Peak peat locality, it would be good to add that to the figure caption.

Figure 2E: The figure caption implies that there are two arrows in the figure, but there is only one in the image.

Line 70: I would replaced "produced" with "discovered" or something similar. "Produced" implies that you are creating the evidence of cyanobacteria when that is not the case.

Line 202: I wonder if there is a better way to indicate the range of sheath widths for Palaeolyngbya kerpii. At first read, I though this was an arrow and checked the reference to make sure it was a greater than symbol. Perhaps it will look better when typeset for the journal, but it may be more readable with a space after the en-dash if the editors approve.

Lines 225-228: This sentence reads a little awkwardly and could be improved.

Lines 63-64: Central should not be capitalized in the locality description. The authors did this correctly elsewhere in the manuscript. It also seems unnecessary to give the CTAM abbreviation since it is not used in any other place in the manuscript.

Line 26: Change "and" to "or"

Line 33: Instead of "As to whether", just begin the sentence with "Whether"

Line 50: Remove "strikingly"

Line 72: Remove "very"

Line 260: I would say "were destroyed"

Line 263: Remove the comma

Experimental design

As this is a descriptive paper, there is no traditional experimental design or research question. The significance of the discovery of this cyanobacterium and how it fills a knowledge gap in the ecology of organisms preserved in permineralized Mesozoic peat is stated by the authors. The methods used to prepare and image the specimens are both appropriate and standard for microorganisms preserved in this manner. The preparation methods are described in enough detail that anyone with similarly preserved fossils would be able to prepare and image their specimens in the same manner.

Validity of the findings

The results, in terms of the description of the cyanobacterium, are well done and thorough. When the authors speculate about certain structures, the preservation of the fossils, or the place of the cyanobacterium within the ecosystem, it is clearly identified and certainly within reason. The discussion and conclusion sections clearly link back to features described in the results.

Lines 262-267: I'm not sure that those studies of cyanobacterial biomass in modern peat ecosystems support the idea that cyanobacteria were rare in the Fremouw ecosystem. Even if cyanobacteria were a small percentage of the total biomass, it doesn't explain why no other algae or bacteria have been described from the locality. If cyanobacteria were only 4% of the total biomass, that still represents a large population of cyanobacteria that is not seen in the peat, without even considering the lack of fossils from algae or other bacteria. These lines seem like an unnecessary critique of the "cyanobacteria everywhere hypothesis". It would be a more valid critique if there were other microorganisms found throughout the peat.

---

## Round 0.2 · accepted · Accept

This revised manuscript has now been accepted and represents a novel finding from the Triassic of Antarctica. The authors have effectively responded to all the reviewer comments and adjusted their manuscript accordingly.

Reviewer 1 ·

Basic reporting

This revised manuscript, documenting the first Mesozoic cyanobacterial fossils preserved in fungal hyphal masses within a leaf-rich permineralized (silicified) peat, is well written, succinct and represents a novel finding from the Triassic of Antarctica. The authors have effectively responded to all the reviewer comments and adjusted their manuscript accordingly. I can find no additional points of disagreement or need for elaboration of contraction of the text. Enlargement of Fig. 2C clarifies the morphology of the fungal masses. The references are appropriate and the illustrations effectively convey the message of the text. The only changes I would make are entirely cosmetic and, in some cases reflect personal grammatical preferences. They include:
Line 80. I would change “…described in Harper et al…” to “described by Harper et al.” Saying “described in” implies it was described inside the authors.
Line 83-84. I would hyphenate “low-sinuosity” as this is used as a combined adjectival modifier for braided streams.
Line 85. I would change the position of the hyphen to “trough cross-bedded”
Line 231. Change “fossil” to “fossils”
Line 288. I suggest adding a comma either side of “therefore”

Experimental design

The research is within the scope of the journal. The research objectives are clearly outlined and appropriately investigated.

Validity of the findings

The results are clear, well presented and the interpretations are supported as far as possible given the constraints of the fossil preservation.

Additional comments

This is a well-presented paper and makes a solid contribution to Antarctic palaeobiology.

Reviewer 2 ·

Basic reporting

The authors have made all of the requested changes that improve the clarity of the language, improve the clarity of figures, and have also added additional literature references.

Experimental design

No comment.

Validity of the findings

The authors have adequately addressed all reviewer comments related to their findings and interpretations.